# Sustainability of Small-Scale Forestry and Its Influencing Factors in Lithuania

**Stasys Mizaras, Asta Doftartė \*, Diana Lukminė and Rita Šilingienė**

Instituto al. 1, Lithuanian Research Center for Agriculture and Forestry, Kėdainių distr., LT-58344 Akademija, Lithuania; stasys.mizaras@gmail.com (S.M.); diana.lukmine@lammc.lt (D.L.); ritasilingiene@gmail.com (R.Š.)

\* Correspondence: asta.doftarte@gmail.com

**Abstract:** Small-scale private forestry is widespread in many countries and occupies 40.3% of the total forest area in Lithuania. The pursuit of sustainability has become one of the main goals of forest policy. In order for small-scale private forestry to be based upon sustainability principles, its sustainability must first be assessed and analyzed. This study assesses the sustainability of 385 small forest holdings of Lithuania using established forest sustainability assessment methods and performs an analysis of the factors influencing the sustainability of small forest holdings using correlation analysis. The Lithuanian small-scale forest holdings were categorized in terms of their level of sustainability as being very high and high (assessed on a five-point scale as 3.5–5 points)—13.6%, middle (2.5–3.5 points)—28.8%, or low and very low (1.0–2.5 points)—57.6%, with the corresponding proportion of holdings indicated as a percentage. A total of 40 independent variables were hypothesized, and their correlation with the sustainability assessments of the holdings was verified. The correlation analysis found mostly weak but reliable ($p < 0.05$) relationships with 23 independent variables: very weak—12 variables, weak—7 variables, middle—2 variables, and strong—2 variables. Moderate and strong correlations were found for the following variables: the owner's view of the forest's economic importance (correlation coefficient: 0.862), income per hectare (0.840), the importance of forestry in the common activity of the owners (0.525), the percentage of mature stands (0.476), the diversity of activities in forest holdings (0.361), and how the wood is used (0.328).

**Keywords:** sustainability; small-scale forestry; holding; correlation analysis

## 1. Introduction

The small-scale forest sector is booming in many countries and plays a crucial role in their rural economies. However, much of the activity in this sector is informal, owing to poorly designed policy frameworks and the lack of political support for small-scale operators. Hoare noted that the development of a framework of indicators could provide an effective way of galvanizing political support and of driving progress towards establishing a legal and sustainable small-scale forest sector [1]. Moreover, sustainability assessments are usually conducted for supporting decision-making and policy development in a broad context. Indeed, assessing sustainability is increasingly becoming common practice in product, policy, and institutional appraisals. The pursuit of sustainability has become one of the main goals of forest policy in many countries, including Lithuania. In order for forestry to be based upon sustainability principles, it must first be assessed and analyzed.

The United Nations Conference held in Rio de Janeiro (1992) defined forest management sustainability as follows: Forest resources and forest lands should be sustainably managed to meet the social, economic, ecological, cultural, and spiritual needs of present and future generations [2]. In this study, the terms "forestry sustainability" and "forest management sustainability" are used synonymously. Sustainable forest management incorporates three pillars: economic, environmental,

and sociocultural [3]. Moreover, these three pillars of sustainability are supplemented with seven thematic elements of sustainable forest management: the extent of forest resources, the biological diversity of the forest, forest health and vitality, the productive functions of forest resources, the protective functions of forest resources, the socio-economic functions of forests, and policy and institutional frameworks [4]. Each thematic element (criterion) is described by indicators. Currently, 11 intergovernmental, regional, and international forest-related processes use sustainable forest management criteria and indicators [5]. In the context of sustainability assessment, the analyst often needs to combine different methods, models, and indicators [6].

A multicriteria method that assesses the sustainability of forestry is the most developed and applied. The method is based on the following:

(1)  Composing a set of criteria and indicators,
(2)  Evaluation of the importance of the criteria and indicators, and
(3)  Procedure for evaluation [7].

By using this scheme, assessments of forestry sustainability have been carried out in different countries [7–17]. Research on the assessment of forestry sustainability has been summarized in various evaluation guidelines [18,19].

The complexity and the multidimensional facets of sustainable development are pushing the scientific community to find new models and paradigms, leading, in recent times, to the emerging field of sustainability science [20].

In the literature, there is no study analogous to this combination of all aspects of a multicriteria assessment of forestry sustainability, including the criteria and indicators, an overall sustainability assessment, an assessment of each holding according to its attribute, and the application of a large number of holdings to the analysis of factors influencing the sustainability of forestry. In measuring sustainable forest management in Tiera del Fuego, Argentina [10], and in the private forests of Halliburton Forest and Wild Life Reserve in Ontario, Canada [11], the authors included all stages of the multicriteria method for the assessment of forest management sustainability: the identification of criteria and indicators, the assessment of the relative importance of criteria and indicators, scores indicating the degree of sustainability, and an overall sustainability point calculation. However, one large holding was assessed without the ability to analyze the effects of the factors on sustainability. By evaluating the level of sustainability of a privately managed forest in Bogor, Indonesia [15], and assessing community forest management [17] and factors influencing sustainability in Iran [13,14], the level of sustainability for each indicator is evaluated by experts rather than calculated by holding attributes.

At a national level, Lithuania participates in the Pan-European Forest Process for the periodic assessment of forest management sustainability conducted by the Ministerial Conference on the Protection of Forests in Europe [21–24]. Parts of Lithuanian forests are private, and this has been the case since 1991 when forests were returned to their former owners or heirs. In 2018, there were 250,000 forest owners (including individual farmers, families, limited companies, and cooperation agreements between these) and 854,200 ha of private forests registered in Lithuania. On average, one forest owner owns 3.4 ha of forest [25].

Small-scale private forestry occupies 40.3% of the total forest area in Lithuania. In order to base small-scale private forestry on sustainability principles, it is helpful to assess and analyze sustainability, specifically in the case of small-scale forestry, given its peculiarities. For assessing small-scale forestry sustainability, the harmonized method should be developed and tested. In Lithuania, the sustainability of small-scale private forestry has not yet been evaluated.

The aim of this study is to assess the sustainability of Lithuanian small-scale forestry and its influencing factors. The methodology of such a study is based on sustainability assessments of forest holdings using established forestry sustainability assessment methods.

In this study, established forestry sustainability assessment methods [18,19] were adopted for the sustainability assessment of Lithuanian small-scale forestry holdings. After assessing the

forestry sustainability of 385 randomly selected small-scale holdings, the distribution of Lithuanian small-scale forest holdings according to the level of sustainability on a five-point scale was determined. Correlation analysis was applied for the evaluation of forestry sustainability relationships with independent variables.

## 2. Materials and Methods

This study involved the following three steps: (1) obtaining a representative sample of small-scale private forest holdings and surveying them, (2) assessment of the forestry sustainability of each holding, and (3) correlation analysis of the factors affecting sustainability.

### 2.1. Samples and Survey

The sample size for the study (n) was calculated using the following formula [26]:

$$n = \frac{N \cdot 1.96^2 \cdot p \cdot q}{\varepsilon^2 \cdot (N-1) + 1.96^2 \cdot p \cdot q} \tag{1}$$

where N is the population size, the value 1.96 corresponds to a 95% confidence level of the standard normal distribution, $p$ is the predicted probability that the analyzed attribute will be evident in the surveyed population (usually the probability of observing the worst scenario, this attribute is typical for half of the population, $p = 0.5$), $q$ is the probability that the analyzed attribute will be evident in the surveyed population ($q = 1 - p = 0.5$), and $\varepsilon$ is the required accuracy, usually $\varepsilon = 0.05$.

A sample size of 385 holdings corresponded to a sampling error of 5%, with a confidence level of 95%. The required number of holdings was randomly selected from the Database of Forest Holdings of the Lithuanian Register Center (further referred to as "Database"). Data for each forest holding needed to be assessed, an analysis of the sustainability of the forestry was determined on the basis of the Database, and the forest owner questionnaire survey was conducted using the face-to-face method. This method is based on direct communication with the respondent in accordance with the prepared questionnaire. Face-to-face interviews help with more accurate screening and allow for a gathering of more and deeper information about the sustainability of small-scale forestry. This method has disadvantages, however, including a comparatively high cost, a time-consuming survey process, the requirement of highly skilled interviewers, and manual data entry.

Forest owners responded to questions describing the factors influencing the sustainability of forestry. All factors were divided into three groups: factors describing the forest owner, the forest holding, and forest management activities. Forest owner indicators included the following: age (years), gender (male, female), education (primary, secondary, higher), forestry knowledge (higher forestry education, experiences in forestry, courses and seminars, mass media, lack of forestry knowledge), the importance of forest function for the forest owner (economic, ecological, and social), residence (in a city or in a village), and the forest management objectives of the forest owner (sale of the holding, sale of the holding after logging, revenue from timber sales, wood for own use, forest for recreation, investment in the forest, the protection of nature and biodiversity, the use of non-timber wood forest products, hunting, and handing the forest over to heirs). Forest holding indicators included the following: size (ha), the distance between forest owner holding and residence (km), the percentage of mature stands, compactness (holding—in one or in two or more areas), the percentage of commercial forest, the percentage of protected forest, and the percentage of recreational forest. Forest property and forest owner activity indicators included the following: how the forest is acquired (not purchased or purchased), prospects for holding usage (will sell, will sell part of it, will pass it on to heirs, will buy more forest, and will plant new forest), forest income per ha (EUR/ha), visits to the forest (one or more times per week, 1–2 times per month, 1–2 times per 6 months, 1–2 times per year, or 1–2 times per 3 years), the importance of forest revenue for the forest owners (basic income, constant additional income, income is rare, no received income, or expenses only), how the wood is used (only for own use, mostly

for own use and the rest sold, most sold and the rest is for own use, everything is sold), the diversity of activities in forest holdings (the number of activities, e.g., cutting, thinning, and planting in the forest), the importance of the forestry in common activities (basic activity, additional, or episodic), the average income of the holding's owner (EUR/month), views on private forest ownership (strong or minimal state regulation and control), the kinds of property rights (personal or common), and affiliations with organizations (member of an organization of forest owners, member of other organizations, or none). The value of these indicators is evaluated as a rational number (e.g., the size of holding), as a dichotomous indicator (e.g., gender), or as an ordinal n-point scale.

## *2.2. Assessment of Sustainability*

The methodology of the forestry sustainability assessment is based on a multicriteria model that assesses sustainability in forestry, which consists of three stages: Stage 1 involves the development of the indicators, Stage 2 involves the estimation of the relative importance of the criteria and indicators, and Stage 3 involves the assessment of each criterion and its indicators [7].

### 2.2.1. Criteria and Indicators

The assessment of Lithuanian small-scale private forestry sustainability is based on European criteria and indicators [21–24]. The assessment of small private forest holdings in Lithuania by each of the 35 European indicators is problematic due to information provision. Many indicators, such as the percentage of natural ecosystem areas at risk of eutrophication for an emission scenario based on current legislation, the C/N index, i.e., the median value for the country, the percentage of sample trees in defoliation classes 2 + 3 + 4, and the landscape pattern index, are not set for Lithuanian private forests and should not be used as indicators to assess the sustainability of private forests. In this study, we followed suggestions [27,28] for selecting key indicators (6–10) when assessing the sustainability of forestry, and these indicators focus on the most essential aspects of sustainable forest management. Six indicators were used in this study and were selected according to the possibilities of information provision in each holding, namely, two for each economic, ecological, and social criterion. The economic criterion is represented by two indicators: (1.1) annual income per hectare, and (1.2) the ratio value of income/growing stock. The estimation of the actual values of Indicator 1.1 is based on the allowable final cut in holdings and the average stumpage prices. The allowable final cut is calculated by dividing each holding volume of mature stands per 1 ha by 10 years and multiplying that by the coefficient of merchantable wood (0.84). The forest stumpage price was the average roundwood market prices of the year 2018 in Lithuanian private forests (39.0 EUR/m$^3$) minus the average logging costs differentiated by holding size. According to a survey of private forest owners, 32.6% of forest holdings do not use their forests and generate no income from them. It is important to assess the annual income of each forest owner. The value for Indicator 1.2 is determined by dividing the values of Indicator 1.1 by the volume of stands per hectare (in 1000 m$^3$). The ecological criterion is represented by Indicators 2.1, the share of protected and protective forests, and 2.2, carbon in stands. Indicator 2.1 is determined for every forest holding and depends on the percentage of protected and protective forests in relation to the total area of the holding. Indicator 2.2 is calculated according to the guidelines of the Global Forest Resources Assessment [29] by multiplying each holding stand's volume per hectare by the basic wood density factor (0.53) and the carbon factor (0.50). The social criterion is represented by Indicators 3.1, the annual number of working days in the forests, and 3.2, the share of recreational forests. Indicator 3.1 was calculated on the basis of multiplying the annual cutting (Indicator 1.1) by the working time per 1 m$^3$ (an average of 0.34 days in Lithuanian private forestry in 2018) and dividing that by the holding area. Indicator 3.2 was estimated on the basis of the percentage of the recreational forest area of each holding in relation to the total area of the holding.

The actual values of the indicators were evaluated as a scale of threshold values (Table 1). The scale is based on the assumption that the mean value of indicators corresponds to the mean of the scales—3.0 points.

**Table 1.** The threshold values for the assessment of Lithuanian private forestry sustainability.

| Criteria and Indicators | Unit | Points and Threshold Values | | | | |
|---|---|---|---|---|---|---|
| | | 1 | 2 | 3 | 4 | 5 |
| **1. Economic** | | | | | | |
| 1.1. Annual income per hectare | EUR/ha | <45 | 45–75 | 75–105 | 105–135 | >135 |
| 1.2. Ratio value of income/growing stock | EUR/1000 m$^3$ | <165 | 165–275 | 275–385 | 385–495 | >495 |
| **2. Ecological** | | | | | | |
| 2.1. Share of protected and protective forests | % | <8 | 8–16 | 16–33 | 33–41 | >41 |
| 2.2. Carbon in stands | t/ha | <22 | 22–45 | 45–91 | 91–114 | >114 |
| **3. Social** | | | | | | |
| 3.1. Annual number of working days in the forests | day/ha | <0.6 | 0.6–1.0 | 1.0–1.4 | 1.4–1.8 | >1.8 |
| 3.2. Share of recreational forests | % | <0.5 | 0.5–0.9 | 0.9–1.3 | 1.3–1.7 | >1.7 |

### 2.2.2. Assessment of Importance

An expert survey was conducted to clarify the importance of the criteria and indicators that describe the sustainability of private forestry in Lithuania. A team of specialists working in the fields of forestry (68%) and environmental protection (32%) was assembled. The majority (73%) of the experts had a master's or doctorate degree, and 27% had a bachelor's university education. The size of the expert group was determined based on the idea [30] that the optimal number of experts should be from 10 to 18. For each survey level (economic, ecological, social), 10 experts were interviewed. The 30 experts evaluated each of the criteria and indicators. The rating method used in reference [8] was used for an assessment of the importance of the criteria. The rating method directly assigns weights (points) explicitly to each criterion by distributing 100 points (percent), which is the sum of all the weights.

Ranking methods [8] were used for the importance assessment of the indicators. The ranking assessment involves an analysis in which each indicator is assigned a "rank" depending on its perceived importance. Ranks were assigned following a 10-point scale (from 1—not very important, to 10—extremely important). The relative weight for indicators is calculated as follows [8]:

$$w_{ji} = \frac{\sum_k r_{jki}}{\sum_i \sum_k r_{jki}} \tag{2}$$

where $w_{ji}$ is the relative weight of indicator $i$ in criterion $j$, and $r_{jki}$ is the rank of indicator $i$ in criterion $j$ by expert $k$.

### 2.2.3. Model

The model used for the assessment of forestry sustainability for each holding is based on references [8,10,31]:

$$S = \sum_j \sum_i (S_{ji} \cdot w_{ji} \cdot w_j) \tag{3}$$

where $S$ is the overall holding sustainability point, which is the relative weight of criterion $j$, $w_{ji}$ is the relative weight of indicator $i$ in criterion $j$, and $S_{ji}$ is the individual sustainability point threshold value for indicator $i$ in criterion $j$.

### 2.3. Factors Analysis

Based on the questionnaire survey, we assessed forestry sustainability (Table 2), hypothesized the number of factors (independent variables) that impact the sustainability of small-scale forestry, and characterized forest owners, forest holdings, property rights, and activities. The dependent variable is the overall sustainability point calculated for each of the 385 holdings. Correlations between sustainability and independent variables were estimated using SPSS for data analyses and analysis results are presented in Table 3.

**Table 2.** Level of holdings' sustainability.

| Sustainability Level | Points | Number of Holdings | % |
|---|---|---|---|
| Very high | 4.5–5.0 | 1 | 0.3 |
| High | 3.5–4.5 | 51 | 13.2 |
| Middle | 2.5–3.5 | 111 | 28.8 |
| Low | 1.5–2.5 | 137 | 35.6 |
| Very low | 1.5–1.0 | 85 | 22.1 |
| Total | | 385 | 100.0 |

**Table 3.** Description of independent variables. SD = standard deviation.

| Variables | Description | Mean | SD |
|---|---|---|---|
| **Forest owners** | | | |
| Age | Age of forest owners (years) | 62.23 | 14.72 |
| 1. Gender | 1—male; 2—female | 1.50 | 0.50 |
| 2. Education | 1—primary, secondary; 2—higher | 1.24 | 0.43 |
| Forestry knowledge | 5—higher education; 4—experiences in forestry; 3—courses and seminars; 2—mass media; 1—no knowledge. | 2.32 | 1.06 |
| Residence | 1—residing in a city; 2—residing in a village | 1.61 | 0.49 |
| Importance of forest functions: | 5—highly important; 4—important; 3—no opinion; 2—not important; 1—not important at all | | |
| Economical | | 4.02 | 1.26 |
| Ecological | | 4.10 | 1.31 |
| Social | | 3.52 | 1.53 |
| Importance of objectives for forest owners: | 5—highly important; 4—important; 3—no opinion; 2—not important; 1—not important at all | | |
| Forest activity | | 1.35 | 1.09 |
| Sale of the holding | | 1.35 | 1.10 |
| Sale of the holding after logging | | 1.08 | 0.50 |
| Revenue from timber sales | | 1.70 | 1.42 |
| Wood for own use | | 3.11 | 1.91 |
| Forest for recreation | | 1.31 | 0.99 |
| Investing in forest | | 1.72 | 1.50 |
| Protection of nature and biodiversity | | 1.70 | 1.44 |
| Use of nonwood forest products | | 1.76 | 1.47 |
| Hunting | | 1.17 | 0.73 |
| Handing down to heirs | | 2.76 | 1.87 |
| **Holdings** | | | |
| Size | Size of holdings in ha | 18.94 | 55.48 |
| Distance | Distance between forest and the residence (km) | 26.42 | 40.93 |
| Mature stands | % of mature stands | 37.61 | 35.41 |
| Compactness | 1—holdings are in one area; 2—holdings are in two or more areas | 1.32 | 0.49 |
| Commercial forest | % of commercial forest | 69.64 | 41.03 |
| Protective forest | % of protective forest | 28.71 | 40.06 |
| Recreational forest | % of recreational forest | 1.39 | 10.63 |

**Table 3.** *Cont.*

| Variables | Description | Mean | SD |
|---|---|---|---|
| **Property and activity** | | | |
| How the forest was acquired | 1—not purchased (restituted, inherited, donated); 2—purchased | 1.22 | 0.41 |
| Prospects for holding usage | 1—will sell; 2—will sell some; 3—will pass it on to heirs; 4—will buy more forest; 5—will plant new forest | 2.91 | 0.70 |
| Income per hectare | Stumpage prices of annual cuttings (EUR/ha) | 74.07 | 97.72 |
| Visits to forest | 5—one or more times per week; 4—1–2 times per month; 3—1–2 times per half a year; 2—1–2 times per year; 1—1–2 times per 3 years | 3.15 | 1.77 |
| Knowledge of boundary | 1—if answered "yes"; 2—no | 1.13 | 0.34 |
| Importance of forest revenue | 5—answered "basic income"; 4—constant additional income; 3—income is rare; 2—does not receive income; 1—costs only | 2.29 | 0.79 |
| How the wood is used | 1—only for own use; 2—mostly for own use, and the rest sells; 3—most sells, and the rest is for own use; 4—everything sells | 2.70 | 1.80 |
| Diversity of activities in forest holdings | Number of activities | 3.01 | 2.74 |
| Importance of forestry in common activity | 4—basic activity; 3—additional; 2—episodic; 1—no activity | 1.90 | 0.78 |
| Total income | Average common income of holdings owner, EUR/month | 346.84 | 460.93 |
| Property right | 1—personal property; 2—common property | 1.55 | 0.80 |
| Affiliation with organization | 3—forest owners' organization; 2—other; 1—none | 1.17 | 0.56 |
| View to private forest ownership: | 1—I completely agree; 2—I agree; 3—I am not making a decision; 4—I disagree; 5—I completely disagree | | |
| Strong state regulation and control | | 3.55 | 1.31 |
| Minimal state regulations and control | | 4.34 | 0.92 |

## 3. Results and Discussion

### 3.1. Ratings and Rankings

The rating method was used for the valuation of the importance of forestry sustainability criteria: economic—42.7%, ecological—32.1%, and social—25.2%.

The importance of forestry sustainability indicators was determined using the rating method. On a 10-point scale, the significance of Indicator 1.1 was rated at 8.035, Indicator 1.2 at 6.207, Indicator 2.1 at 7.133, Indicator 2.2 at 5.724, Indicator 3.1 at 8.033, and Indicator 3.2 at 6.933.

### 3.2. Holdings' Sustainability

The forestry sustainability of 385 holdings was assessed. Low sustainability holdings (35.6%) prevailed over very high and high (13.5%), middle (28.8%), and very low (22.1%) sustainability holdings (Table 2). The average was 2.39 points.

### 3.3. Correlation Factors

Correlation analysis of forest holding sustainability scores and its influencing factors (Table 3) was performed. The characteristics of the indicators, which have been used for modeling the influence on the sustainability of forestry, are expressed in numbers. Three different types of values are used to show the data sizes or values. Some indicators, such as age, forest holding size, and distance, were directly expressed in numbers. The other part of the data (dichotomous indicators) was classified and expressed as 1 or 2, such as gender and education. The third part shows numbers of the ordinal n-point scale, such as the importance of objectives for forest owners and the prospects for holding usage. Based on the numerical values of these variables, the mean values of the variables and the standard deviation (SD) were determined. Table 3 (forest owners) reports the socio-demographics of the analyzed sample. Slightly more than half (50.3%) of the respondents were women, and the average age of respondents was 62 years. About 24% of the respondents had higher education. According to the level of forestry knowledge, the respondents are distributed as follows: 10.2% had working experience in forestry,

8.1% graduated in forestry studies, 15.9% participated in courses and seminars, and 27.6% had no knowledge of forestry. Sixty-one percent of the respondents lived in rural areas.

Respondents indicated the relative importance of the three main forest functions: economic (5—very important (50.8%), 4—important (22.4%), 3—neither yes nor no (12.8%), 2—not important (6.3%), 1—not important at all (7.8%)), ecological (5—very important (58.7%), 4—important (15.7%), 3—neither yes nor no (11.7%), 2—not important (4.4%), 1—not important at all (9.4%)), and social (5—very important (38.5%), 4—important (20.6%), 3—neither yes nor no (13%), 2—not important (9.9%), 1—not important at all (18%)). The most important objectives for forest owners were using wood for personal use (45.1% of the respondents indicated that it is very important) and handing it down to their heirs (33.1% of the respondents indicated that it is very important).

The second part, "Holdings," describes the forestry holdings. The average size of respondents' holdings is 18.94 ha, and the average distance between the forest and the living residence is 26.42 km. The characteristics of forest stands are as follows: almost 38% of the forest stands area is mature, 69.64% is commercial, 28.71% is protected, and 1.39% is recreational. About two-thirds (67.2%) of forest holdings are located on one compact piece of land. Three main types of forest holding acquisition (restitution, inheritance, and donation) were predominant among respondents. The forest holdings resituated, inherited, or donated were reported by 78.1% of respondents. Further disposal of forest holdings was reported as follows: 11.4% of respondents plan to sell all or some forest holdings, 77.1% will pass it on to heirs, and 11.5% plan to expand their forest holdings. Most forest owners (84.4%) know the boundaries of their forest holdings, but there are owners (15.6%) who do not even know where their forest is located or are not sure about their forest holding boundaries. About one-fourth (25.3%) of respondents reported that they visited a forest holding on a weekly basis during the last 12 months prior to the interview, 21.1% went to a forest one to two times a month, and 53.4% stated that they visited a forest less than once a month or not at all during that time period. The average income per hectare was 74.07 (EUR/ha), and the average common income of holding owners was 346.84 EUR/month.

Most respondents (78.8%) incur costs or do not have any income from the forest holding, some of them have basic or additional income (7.3%), and 14.1% rarely receive income.

Respondents reported how they use their wood from their forest holding: only for own use—70.7%, mostly for own use and the rest sold—9.3%, most sold and the rest is for own use—10.2%, or everything is sold—9.8%.

We can divide the interviewed forest owners into several groups according to the intensity of forest management: very active (31.5%), active (51.3%), passive (12.8%), and private forest owners who had no management activity in the forest holding (4.4%). Respondents indicated how many different activities they carried out with their forest holding, and the average number was 3.01.

Forest owners' attitudes towards private forest ownership and state regulation differed: 22.9% of respondents believed that strong regulation and forest management control are needed from state authorities, 17.7% said neither yes nor no, and 59.4% believed that minimal state regulations and control are needed.

About one-third (34.9%) of respondents manage forest holdings together with co-owners, and for 65.1% of respondents, the forest holding is personal property.

Respondents indicated membership with different organizations: 2.6% are members of organizations representing forest owners (forest owners associations), 12.0% are members of other organizations not related to forestry or the environment, and 85.4% do not have memberships with any organizations.

Our correlation analysis found mostly weak but reliable ($p < 0.05$) relationships with 23 independent variables (Table 4). Moderate and strong correlations were associated with the following variables: the owner's view of the forest's economic importance (0.862), income per hectare (0.840), the importance of forestry in the common activity of the owners (0.525), the percentage of mature stands (0.476), the diversity of activities in forest holdings (0.361), and how the wood is used (0.328).

**Table 4.** Correlations between holdings' sustainability and independent variables.

| Independent Variables | Coefficient of Correlation | *p* |
|---|---|---|
| Gender | 0.125 * | 0.01 |
| Forestry knowledge | 0.181 ** | 0.000 |
| Economic importance of forest functions | 0.862 ** | 0.000 |
| Residence | 0.132 ** | 0.01 |
| Importance of Objectives | | |
| Forest activity | 0.108 * | 0.034 |
| Revenue from timber sales | 0.187 ** | 0.000 |
| Wood for own use | 0.183 ** | 0.000 |
| Use of non-wood forest products | 0.112 * | 0.028 |
| Hunting | 0.170 | 0.001 |
| Handing over to heirs | 0.218 ** | 0.000 |
| Size | 0.249 ** | 0.000 |
| Mature stands | 0.476 ** | 0.000 |
| Compactness | 0.219 ** | 0.000 |
| How the forest was acquired | 0.202 ** | 0.000 |
| Prospects for holding usage | 0.188 ** | 0.000 |
| Income per hectare | 0.840 ** | 0.000 |
| Visits to forest | 0.108 * | 0.030 |
| Knowledge of boundary | 0.146 ** | 0.004 |
| Importance of forest revenue | 0.203 ** | 0.000 |
| How the wood is used | 0.328 ** | 0.000 |
| Diversity of activities in forest holdings | 0.361 ** | 0.000 |
| Importance of forestry in common activity | 0.525 ** | 0.000 |
| Total income | 0.120 * | 0.02 |

* Significant at 10% level; ** significant at 5% level.

## 3.4. Discussion

The principles and methods for assessing the sustainability of forest management, developed in Europe and elsewhere in the world, can be applied to assess the sustainability of private forestry in Lithuania. The study results indicate the sustainability of Lithuanian small-scale forestry and its peculiarities. The 2.39-point result of the average sustainability of selected forest holdings indicates a low level of sustainability. It is lower than that assessed for the sustainability of the total Lithuanian forestry, which was 3.31 points [23].

The lowest scored indicator was 3.2—share of recreational forests—with an average of 1.14 points, and the highest scored indicator was carbon in stands with 3.06 points. Indicator 1.1, annual income per hectare, was assessed as 2.41 points, Indicator 1.2, the ratio value of income/growing stock, at 2.48 points, Indicator 2.1, the share of protected and protective forests, at 2.40 points, and Indicator 3.1, the annual number of working days in the forests, at 2.65 points. The evaluation of forestry sustainability is determined by different factors. A reliable but weak correlation was identified for many factors. This was also found in a study of factors affecting sustainable forest management performed in Iran [13]. However, the set of factors in this survey is different.

It was found that the strongest influencing factors of Lithuanian small-scale forestry are the owner's view of the forest's economic importance (correlation coefficient: 0.862), income per hectare (0.840), the importance of forestry in the common activity of the owners (0.525), the percentage of mature stands (0.476), the diversity of activities in forest holdings (0.361), and how the wood is used (0.328). A correlation coefficient measures the strength and direction between two variables. The scales of positive correlation coefficient values are as follows: very weak is represented by a range from 0 to 0.2, weak from 0.2 to 0.5, average from 0.5 to 0.7, strong from 0.7 to 1, and very strong is represented by 1 [26]. The correlation coefficient shows the existence of the relationship and its strength. The strength of the correlation shows the influence of the variable on the assessment of sustainability.

The information obtained from this study is relevant for stakeholders involved in the formation and implementation of the Lithuanian private forestry policy. Many problems were encountered in assessing the sustainability of the Lithuanian private forestry. The Lithuanian National Forest Strategy does not contain strategic objectives and indicators describing private forestry that could facilitate the creation of the list of criteria and indicators for assessing the sustainability of forestry. There is no systematic monitoring of the objectives and problems experienced by private forest owners in Lithuania. Another problem with assessing the sustainability of Lithuanian private forestry is the lack of data. Even if Lithuanian forestry statistics contain considerable data on general wood resources and their use, data on the financial, ecological, and social indicators of private forestry are still lacking.

Due to information provision problems, the application of the full list of European criteria and indicators for the assessment of the sustainability of small-scale forestry in Lithuania is complicated, and the option of key indicators has to be applied. However, as our previous studies on the impact of the number of indicators on the assessment of forestry sustainability have shown [32], due to the use of a range of from 1 to 9 indicators for each criterion (economic, ecological, and social) in assessment scenarios, the assessments differ slightly, from 3.03 to 3.46 points on a five-point scale. Further research and activities will be required in the future, both to support the target indicators and for the development of private forestry statistics.

## 4. Conclusions

The sustainability of small-scale private forest holdings has neither been assessed nor analyzed in Lithuania. The principles and methods for assessing the sustainability of forest management, developed in Europe and elsewhere in the world, can be applied to assess the sustainability of private forestry in Lithuania.

The wide range of data about private forest owners, their properties, and forestry activity were collected during the survey. The results of the study show that a significant proportion of private forest owners are elderly (the average age of respondents was 62 years), and more than half (50.3%) of the respondents are women. Comparing the data with previous surveys of private forest owners, it can be stated that private forest owners in Lithuania are getting younger, but the prevailing trend of female owners remains [33]. About one-third (34.9%) of respondents manage forest holdings together with co-owners. This is also in line with the data provided by the State Forest Service that approximately one-third of forest holdings in Lithuania are managed together with co-owners [26]. About 24% of the respondents had higher education, and a relatively high percentage of private forest owners (72.4%) have different levels of knowledge about forest management, which indicates that forest owners are interested in forest management activities. It assumes that forest holdings will be managed in accordance with at least a minimum level of forest management knowledge. Moreover, 10.2% of forest owners have working experience in forestry. Sixty-one percent of the respondents live in rural areas, and the average distance between the forest and the living residence is 26.4 km. These data show that private forest owners, due to the favorable distance from the place of residence to the forest holding, have the opportunity to take proper and timely care of the forest holding. Near 70% of forest stand areas are commercial; however, about 30% of forest holdings are located in protected areas where forest management activity is limited. About 77% of private forest owners will pass forest holdings on to heirs, which would enable the continuation of forest management goals and the formation of traditional forest owners. However, about 16% of owners do not even know where their forest is located or are not sure about their forest holding boundaries. This data also explains private forest owners' management activity, because nearly 13% of forest owners are passive and 4.4% of private forest owners have no management activity in the forest holding. The characteristics of private forest owners, management practices and traditions, as well as the characteristics of forest holdings, are certainly important in forest policy guideline formation and development program implementation.

After assessing the forestry sustainability of 385 randomly selected small-scale holdings, the distribution regarding the level of sustainability of Lithuanian small-scale holdings as a percentage of

all holdings was determined: very high and high (assessed on a five-point scale, 3.5–5.0 points)—13.5%, middle (2.5–3.5 points)—28.8%, and low and very low (1.0–2.5 points)—57.7%. The average was 2.39 points.

Strong correlations between sustainability of holding and independent variables show how private forest owners should organize forest management activity in order to achieve higher sustainability. Our correlation analysis found mostly weak but reliable ($p < 0.05$) relationships with 23 independent variables: very weak—12 variables, weak—7 variables, middle—2 variables, and strong—2 variables. Moderate and strong correlations were found for the following variables: the owner's view of the forest's economic importance (correlation coefficient: 0.862), income per hectare (0.840), the importance of forestry in the common activity of the owners (0.525), the percentage of mature stands (0.476), the diversity of activities in forest holdings (0.361), and how the wood is used (0.328). The result of correlation analysis showed that the strongest influence on the sustainability of small-scale forestry is economic factors.

Sustainability assessment is usually conducted for supporting decision-making and policy development in a broad context. The wide range of collected data and survey results can be used for formation and implementation programs such as the National Strategy for Sustainable Development, the National Forest Sector Development Program, and other programs related to sustainable development in Lithuania. It can also be used for forestry legal framework regulation and improvement.

**Author Contributions:** This research article is the result of a collaboration of the first contributing authors. These authors contributed to the conceptualization of the paper. S.M. wrote the original draft, and a draft was revised and edited by A.D. and D.L.; R.Š. contributed to the questionnaire survey. All authors have read and agreed to the published version of the manuscript.

**Funding:** This research was supported by the Lithuanian Research Centre for Agriculture and Forestry's long-term program "Sustainable forestry and global changes, 2017–2021".

**Conflicts of Interest:** The authors declare that there are no conflict of interest.

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
