# Peer review of "Sustainability of Small-Scale Forestry and Its Influencing Factors in Lithuania"

_forests, doi:10.3390/f11060619_

Round 1

Reviewer 1 Report

The topic of the paper as well as its results are of importance. Yet, the paper requires significantly more work (particularly in the discussion part) to be ready for publishing. 

Needs intensive work on grammar (e.g., sentences in lines 30, 36 - 37, 40, 47, 48, 55, 59, 61, 77, 80 - and many more)

Introduction: The intro would benefit significantly from a clearer structure (restructure content) and more references, e.g., lines 59-63. Authors constantly switch between information from Lithuania and other countries - better structure of information needed. Line 85: no results in the introduction

Materials and methods: More information on the "face-to-face method" would be beneficial for the readers' understanding of the process (lines 101 to 104). Did you interview 385 forest owners in person? When did the interviews take place? etc.
Irrelevant reference to Table 2 in line 105. 
A better explanation is required for why not all of the 35 European indicators were applied in this study. What is the benefit of excluding 29 indicators? What information is lost due to this approach? Why did you select those six indicators over all others? 
What is the reason for the information in lines 146-147?
Table 2 should be moved to the results and only discussed there, not in the method section.
Weak reasoning for your selection in lines 168-179.
Unclear if "Criteria and indicators" referred to in section 2.2.2 refer to the six selected criteria or the 35 European criteria mentioned before. 
Why did you only select participants with a university degree to evaluate your criteria? Potential selection bias and exclusion of non-degree practitioners.
More precision is needed in the wording throughout the paper - questionnaire survey for example - what does it refer to? The forest owner face-to-face method? 

Results: Lacks the results of the rating and ranking procedure. Results from Table 2 should be integrated in the results section.

Discussion: more effort needs to be put into the discussion. No real discussion is provided. What do the results now mean for Lithuania? The information currently in the discussion should rather be part of the introduction and the reasoning for this study. What is also missing is a concise discussion of the shortcomings of this approach (as outlined before - what is the risk of not applying all 35 criteria). 

Conclusion: 

Reasoning for study weak (lines 65-67)

Author Response

Answers

  1. Needs intensive work on grammar (e.g., sentences in lines 30, 36 - 37, 40, 47, 48, 55, 59, 61, 77, 80 - and many more).

The English will be checked by skilled proofreading providers.  

  1. Introduction:The intro would benefit significantly from a clearer structure (restructure content) and more references, e.g., lines 59-63. Authors constantly switch between information from Lithuania and other countries - better structure of information needed. Line 85: no results in the introduction

Structure of introduction is changed according to the recommendations of reviewer. The introduction has been restructured to separate information from other countries and Lithuania. The source for data in line 59-63 is indicated. Line 85 has been amended to change its meaning.

  1. Materials and methods:More information on the "face-to-face method" would be beneficial for the readers' understanding of the process (lines 101 to 104). Did you interview 385 forest owners in person? When did the interviews take place? etc. 

Description of the method “face-to-face” has been supplemented by a more detailed description. All 385 forest owners were interviewed. We contacted and visited all respondents at their living place. 

  1. Irrelevant reference to Table 2 in line 105. 

Irrelevant reference to Table 2 in line 105 was deleted.

  1. A better explanation is required for why not all of the 35 European indicators were applied in this study. What is the benefit of excluding 29 indicators? What information is lost due to this approach? Why did you select those six indicators over all others?

Due to the information provision problems, the application of the full list of European criteria and indicators for the assessment of the sustainability of small-scale forestry in Lithuania is complicated and the option of key indicators had to be applied. It can be assumed that increasing the number of indicators has weak effect on the overall assessment of sustainability. Although, as our previous studies on the impact of the number of indicators on the assessment of forestry sustainability have shown [32]. Six indicators were selected according to the possibilities of information availability.

  1. What is the reason for the information in lines 146-147?

It is important for assessment annual income of each forest owner.

  1. Table 2 should be moved to the results and only discussed there, not in the method section.

Table 2 is moved to the results chapter.

  1. Weak reasoning for your selection in lines 168-179. Why did you only select participants with a university degree to evaluate your criteria? Potential selection bias and exclusion of non-degree practitioners.

The Delphi sociological expert survey method was used to obtain expert opinions on the importance of criteria and indicators describing the sustainability of small-scale forestry. The expert specially selected persons who have the greatest competence and sufficiently detailed information about the necessity and meaning of the research.

  1. Unclear if "Criteria and indicators" referred to in section 2.2.2 refer to the six selected criteria or the 35 European criteria mentioned before. 

For assessment have been used 3 criteria (economic, ecological and social) and 6 indicators.

  1. Results:Lacks the results of the rating and ranking procedure. Results from Table 2 should be integrated in the results section.

Results have been additionally supplemented by Section 3.1 Rating and ranking. Table 2 integrated to the Results section.

  1. Discussion: more effort needs to be put into the discussion. No real discussion is provided. What do the results now mean for Lithuania? The information currently in the discussion should rather be part of the introduction and the reasoning for this study. What is also missing is a concise discussion of the shortcomings of this approach (as outlined before - what is the risk of not applying all 35 criteria). 

The former part of the discussion has been moved to the introduction part. A new discussion section has been written, highlighting the aspects of the meaning of the results for Lithuania, the application of key indicators and correlation analysis aspects.

  1. Conclusion: Reasoning for study weak (lines 65-67)

Small-scale private forestry occupies 40.3% of the total forest area in Lithuania. In order to base small-scale private forestry on sustainability principles, it is helpful to assess and analyze sustainability specifically in the case of small-scale forestry, given its peculiarities. For assessing small-scale forestry sustainability the harmonized method should be developed and tested. In Lithuania, the sustainability of small-scale private forestry has not yet been evaluated.

Many problems were encountered in assessing the sustainability of Lithuanian private forestry. The Lithuanian National Forest Strategy does not contain strategic objectives and indicators describing private forestry that could facilitate the creation of the list of criteria and indicators for assessing the sustainability of forestry. There is no systematic monitoring of the objectives and problems experienced by private forest owners in Lithuania. Other problem with assessing the sustainability of Lithuanian private forestry is the lack of data. Even if Lithuanian forestry statistics contain considerable data on general wood resources and their use, data on the financial, ecological and social indicators of the private forestry are still lacking.

Reviewer 2 Report

I have been given the opportunity to read an interesting research paper. Some comments:

  • You need to improve your English while writing the paper. It discourages the reader from appreciating your paper.
  • You could improve your introductive section. You quote Iran, but you do not necessarily use the same criteria. Thereby, what is the purpose of listing the influencing factors if you do not test the same factors in your own study?
  • I do not understand the way you interpret the mean values of your independent variables (Table 2). For example, when you read the values of "the importance of forest functions," you obtain scores that normally match with greater sustainability. Nevertheless, this does go along with the scores relative to "the importance of objectives for forest owners." By the way, the description of "view to private forest ownership" lacks the choice values 4 and 5.
  • The content of "Subsection 3.3. Discussion" should rather be in your conclusive section. The second paragraph in the "conclusive section" should rather be in your "discussion."
  • You do not really discuss the results obtained from the correlation analysis. You only recall the moderate and strong correlations. What are the implications of moderate and strong correlations? How do they impact the sustainability of small-scale private forestry? You should answer those questions in "Subsection 3.3."

Author Response

Response to Reviewer

  • You need to improve your English while writing the paper. It discourages the reader from appreciating your paper.

The English will be checked by skilled proofreading providers.

You could improve your introductive section. You quote Iran, but you do not necessarily use the same criteria. Thereby, what is the purpose of listing the influencing factors if you do not test the same factors in your own study?

The reference of Iran survey is deleted from the introduction part.

  • I do not understand the way you interpret the mean values of your independent variables (Table 2). For example, when you read the values of "the importance of forest functions," you obtain scores that normally match with greater sustainability. Nevertheless, this does go along with the scores relative to "the importance of objectives for forest owners." By the way, the description of "view to private forest ownership" lacks the choice values 4 and 5.

The mean value is the average of the values of all 385 forest owners for each indicator. The individual value of each holding was used in the correlation analysis. There was a proofreading error for the indicator „View to private forest ownership“. The error has been corrected.

  • The content of "Subsection 3.3. Discussion" should rather be in your conclusive section. The second paragraph in the "conclusive section" should rather be in your "discussion."

We think that subsection 3.3 Discussion would not fit to the Conclusions section as there are many references in Discussion section. Discussion section have been rewrite.  

  • You do not really discuss the results obtained from the correlation analysis. You only recall the moderate and strong correlations. What are the implications of moderate and strong correlations? How do they impact the sustainability of small-scale private forestry? You should answer those questions in "Subsection 3.3."

Discussion section supplemented additionally by correlation analysis aspects. Correlation coefficient measures the strength and direction between two variables. Scale of positive correlation coefficient values: very weak from 0 to 0.2, weak from 0.2 to 0.5, average from 0.5 to 0.7, strong from 0.7 to 1, and very strong – 1 [26]. The correlation coefficient shows only the existence of relation and its strength, but does not show factors impact to the sustainability of small-scale private forestry.

Round 2

Reviewer 1 Report

Thank you for revising the paper. It improved noticeably from the last version. 
Some issues remain:

Still requires extensive grammatical revisions. 

The introduction improved yet needs grammatical revision - this will improve the flow between the sections, which is currently still missing. In particular, the newly inserted section reads like an introduction on its own and would probably also work as primary paragraph in the introduction. Please check the introduction and its flow one more time and create a more suitable structure (e.g., decreasing repetition).

The methods section became clearer, thank your for adding the required information. The provided reasoning for the exclusion of certain criteria is weak (“are not set for Lithuanian private forests”). Please provide a stronger explanation and revise grammatically so it is clear what you mean.

Table 3 gives a good overview of the results. However, some information in the table is not properly projected. For example providing “mean” values for categorical question such as gender, education, residence, forest acquisition and many more is not statistically appropriate. The paper would benefit from a written explanation of those aspects with suitable depictions of percentages, rather than adding them to the table. I strongly advise to adjust this table and add the necessary text. 
The discussion also improved, yet is still in the shorter side despite interesting results from table 3. 

Author Response

Response to Reviewer 1 Comments

  • Still requires extensive grammatical revisions. 

English language is edited by MDPI. 

  • The introduction improved yet needs grammatical revision - this will improve the flow between the sections, which is currently still missing. In particular, the newly inserted section reads like an introduction on its own and would probably also work as primary paragraph in the introduction. Please check the introduction and its flow one more time and create a more suitable structure (e.g., decreasing repetition).

The introduction is corrected according reviewer comment.

  • The methods section became clearer, thank you for adding the required information. The provided reasoning for the exclusion of certain criteria is weak (“are not set for Lithuanian private forests”). Please provide a stronger explanation and revise grammatically so it is clear what you mean.

The assessment of small private forest holdings in Lithuania by each of the 35 European indicators is problematic due to information provision. Many indicators such as “Percentage of natural ecosystem areas at risk of eutrophication for an emission scenario based on current legislation”, “C/N index, median value for the country”, “Percent of sample trees in defoliation classes 2+3+4”, “Landscape patter index” etc. are not set for Lithuanian private forests and could not be used as indicators to assess the sustainability of private forests.

  • Table 3 gives a good overview of the results. However, some information in the table is not properly projected. For example providing “mean” values for categorical question such as gender, education, residence, forest acquisition and many more is not statistically appropriate. The paper would benefit from a written explanation of those aspects with suitable depictions of percentages, rather than adding them to the table. I strongly advise to adjust this table and add the necessary text. 

Additional information related to the Table 3 is added.

  • The discussion also improved, yet is still in the shorter side despite interesting results from table 3. 

Reviewer 2 Report

Review

  • You should remove the lines corresponding to the deleted table in Subsection 2.3.
  • Instead of recalling the results from your correlation analysis both on pages 9 and 11, you should write about the policy implications of your results in the conclusive section.

Author Response

Response to Reviewer 2 Comments

  • You should remove the lines corresponding to the deleted table in Subsection 2.3.

Additional information is added to Subsection 2.3

  • Instead of recalling the results from your correlation analysis both on pages 9 and 11, you should write about the policy implications of your results in the conclusive section.

Additional information is added to section 4. Conclusions.
